# Predator Diversity Does Not Contribute to Increased Prey Risk: Evidence from a Mesocosm Study

Pierre William Froneman 

SARChI Chair: Marine Ecosystems, Department of Zoology & Entomology, Rhodes University,
P.O. Box 94, Makhanda 6140, South Africa; w.froneman@ru.ac.za

**Abstract:** Predation plays an important organisational role in structuring aquatic communities. Predator diversity can, however, lead to emergent effects in which the outcomes of predator–prey interactions are modified. The importance of predator diversity in regulating predator–prey interactions was investigated during a 9-day mesocosm study conducted in the middle reach of a temporarily open/closed, temperate, southern African estuary. The zooplankton community, comprising almost exclusively (>95% of total counts) calanoid and cyclopoid copepods of the genera *Pseudodiaptomus*, *Paracartia* and *Oithona*, was subject to three different juvenile fish predator treatments at natural densities: 1. predation by *Gilchristella aestuaria*, (Gilchrist, 1913; SL 15.3 ± 2.4 mm); 2. predation by *Myxus capensis* (Valenciennes, 1836; SL 12.8 ± 3.7 mm); and 3. a combination of the two predators. The presence of the predators contributed to a significant decline in the total zooplankton abundances, with a concurrent increase in total chlorophyll-*a* (Chl-*a*) concentrations, consistent with the expectations of a trophic cascade (ANCOVA; $p < 0.05$ in all cases). There were no significant differences in the total Chl-*a* concentration or total zooplankton abundances between the different predator treatments, suggesting that the increase in predator diversity did not contribute to increased prey risk or to the strength of the trophic cascade.

**Keywords:** plankton; predation; juvenile fish; emergent effects; trophic cascade

## 1. Introduction

It is now well established that predation plays an important organisational role in plankton communities and contributes to carbon flow and nutrient cycling within aquatic systems [1–3]. Predators affect not only their prey, but also organisms further down the food web through trophic cascades [4,5]. Predation-induced trophic cascades have been linked to changes in the energy flow and food web structure that ultimately may enable regime shifts within aquatic systems [6].

Food webs comprise numerous species that interact indirectly or directly with one another. Historically, ecologists have assumed that similar predatory species have indistinguishable effects on their prey [7]. Assigning organisms at a similar trophic level to a single ecological unit is unrealistic. The interactions between predatory species (e.g., cooperation, competition and intra-guild predation), as well as anti-predator responses by prey, can lead to so-called emergent 'multiple predator effects' (MPEs), where prey consumption rates by multiple predators can have different outcomes [8–12]. Multiple predators can combine: (1) independently, (2) synergistically [9,13,14] (leading to increased predation) or (3) antagonistically (e.g., leading to prey risk reduction [11]). The emergence of MPEs suggests that predator diversity must be considered when assessing the role of predation in structuring aquatic food webs.

The importance of estuaries as nursery and feeding grounds for fish and invertebrates is well established [15,16]. The coastline of South Africa exhibits turbulent wave action with few sheltered bays. This emphasises the importance of estuaries as nurseries for estuarine, freshwater and marine breeding fish species because they support structurally

complex habitats that provide refugia from predation and exhibit high rates of secondary production [17–19]. Consequently, juvenile fish form an important contribution to the total plankton abundance and biomass and are, therefore, thought to play an important role in ecosystem functioning and the energy dynamics of estuaries within the region [20,21]. Indeed, mesocosm studies conducted in the warm temperate biogeographic zone of South Africa showed that trophic cascades facilitated by predation by early-life-history fish played an important role in determining the plankton biomass and community structure within estuaries [2,3]. Moreover, the feeding activity of early-life-history fish also contributed to the stability within the plankton community [22].

It is estimated that 155 fish species routinely utilise South African estuaries as nurseries [23]. While mesocosm studies have provided strong experimental evidence of the importance of predation by juvenile fish in structuring the plankton community within South African estuaries [2,3], the role of predator diversity in mediating predator–prey outcomes has not been considered. The aim of this investigation was to assess the emergent effects of predator diversity in mediating predator–prey interactions in the plankton community. The study was conducted employing a 9-day mesocosm study in the middle reach of a temperate, temporarily open/closed estuary on the south-eastern coastline of South Africa.

## 2. Materials and Methods

### 2.1. Study Site

The mesocosm experiment was conducted during the closed phase of the temporarily open/closed (TOCE) Kasouga Estuary in the warm temperate zone along the southeast coastline of South Africa over a period of 9 consecutive days (4–15 September 2014; Figure 1). The estuary is approximately 2.5 km in length and is characterised by an extensive salt marsh on the east bank in the lower reaches of the system. Water depths in the estuary typically range from <0.5 m to 1.5 m, depending on the mouth state (open (estuary has breached) vs. closed (presence of a sandbar at the mouth which separates the estuary from the marine environment)) of the system. Surface water temperatures within the estuary range from 14 to 28 °C and demonstrate a strong seasonal pattern with maximum temperatures in summer and minimum in winter [24–26]. Due to the small catchment area (~49 km$^2$) and sporadic rainfall, the system is generally regarded as oligotrophic, with salinities generally >25 (PSU). Following heavy rainfall within the catchment area, the system may become river-dominated, after which time the estuary normally breaches and becomes tidally dominated. Studies conducted within the estuary indicate that the total chlorophyll-*a* (Chl-*a*) concentration is <1.00 μg L$^{-1}$, reflecting the low macronutrient availability as a result of sporadic/reduced freshwater inflow [24]. The zooplankton community is almost entirely (>95%) dominated, both numerically and by biomass, by calanoid and cyclopoid copepods of the genera *Psuedodiaptomus*, *Paracartia* and *Oithona* [2,25]. Total zooplankton abundances within the estuary are highly variable, reflecting changes in food availability and recruitment, and range from $1 \times 10^2$ to $1 \times 10^4$ individuals per cubic metre [24,25]. Like many of the TOCEs within this geographic region, the estuary represents an important nursery for a variety of marine and estuarine breeding fish species [21].

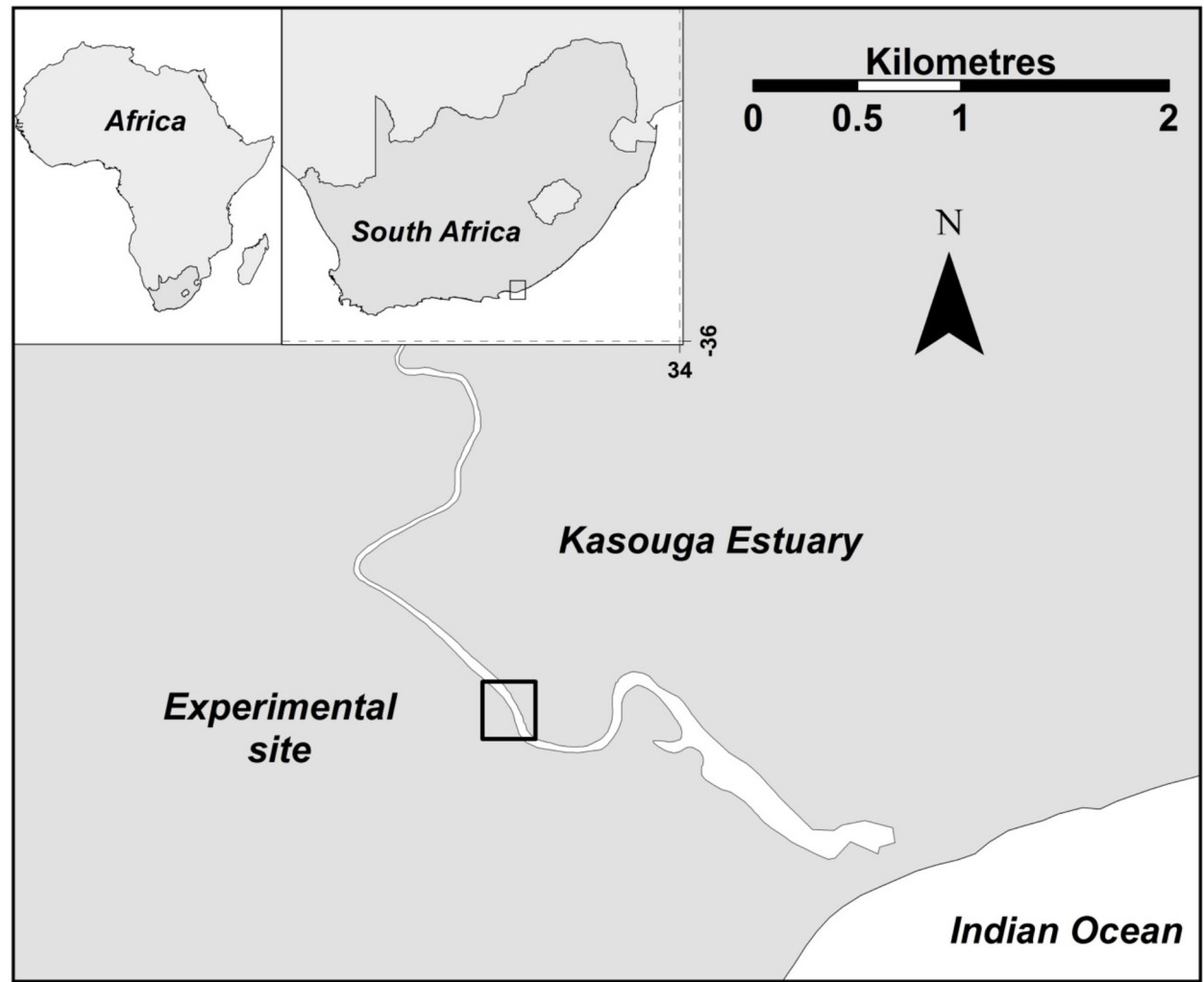

**Figure 1.** Geographic position of the warm temperate, temporarily open/closed Kasouga Estuary, South Africa, showing the location of experimental mesocosm deployments (after Wasserman et al., 2013).

*2.2. Experimental Set-Up*

Fifteen 1000 L mesocosm enclosures (1.4 m deep; 1 m × 1 m) constructed from translucent 200 μm-thick virgin polyethylene bags were established in the middle reach of the estuary in water ~2 m deep, as outlined in Wasserman et al. (2013). Previous studies conducted within the estuary indicate that the system is characterised by the virtual absence of any horizontal gradients in temperature and salinity during the closed phase due to the shallow water depth and strong coastal winds, which facilitate the horizontal mixing of the water column [24–26]. The study site is, therefore, representative of the estuary during the closed phase. Each mesocosm was sealed at the bottom and open to the atmosphere at the top. The tops of the enclosures were secured to a 1 m × 1 m plastic frame fitted with 5 L buoys, elevating the top of the mesocosm 0.5 m above the waterline. This ensured that no overtopping of estuarine water occurred during the study. Each mesocosm was secured to a 30 kg concrete mooring anchored in the estuarine sediment. The top of each mesocosm was fitted with a 4 cm × 4 cm plastic grid to exclude piscivorous birds.

Four trophic treatments and a control (n = 3 for each treatment) were established during the experiment. All experimental manipulations were conducted at night to account for the diel vertical migration patterns exhibited by the zooplankton within the estuary [2,24]. For the control, estuarine water was gravity-fed through a 50 μm mesh to exclude metazoans. Preliminary data indicated that >95% of all metazoans and <5% of

the total Chl-*a* concentration were removed during this procedure (data not shown). In Treatment 2 (zooplankton treatment), estuarine water was gravity-fed through a 750 μm mesh to remove all large predatory zooplankton (e.g., mysids) and juvenile fish. Treatments 3 and 4 comprised water that had been gravity-fed through a 750 μm mesh with a single species of juvenile fish, while Treatment 5 comprised a combination of the two juveniles (designated the Mixed treatment). Early juvenile ($20.7 \pm 2.5$ mm Standard length) freshwater mullet, *Myxus capensis* (Valenciennes, 1836) of the family Mulgilidae (designated the *M. capensis* treatment) and the estuarine round-herring ($13.2 \pm 5.2$ mm Standard length), *Gilchristella aestuaria* (Gilchrist, 1913) of the family Clupeidae (designated *G. aesturia* treatment), stocked at natural densities (determined from seine-net tows conducted 24 h prior to the experiment), were employed as predators. In Treatments 3 and 4, four individuals were added to each mesocosm, while in Treatment 5, two representatives of each species were employed. Fish were captured at the study site 24 h prior to the commencement of the study using a pull-net and were incubated in aerated estuarine water in 300 L containers. Both species are planktivorous during their early life history, feeding mainly on calanoid and cyclopoid copepods within the water column [15]. All the necessary permits for the collection of the zooplankton and juvenile fish in the experiments were obtained from the Department of Agriculture, Forestry and Fisheries (DAFF), Republic of South Africa (permit reference number: RES2011/46).

Measurements of salinity (PSU), temperature (°C) and dissolved oxygen (mg L$^{-1}$) were made using an Aquaread aquameter between 18:00 and 19:00 at a depth of 0.5 m in each mesocosm at the commencement of the study, and every second day thereafter until the end of the experiment. Biological samples were collected shortly thereafter. Total chlorophyll-*a* (Chl-*a*) concentrations within each mesocosm were determined from a 250 mL water sample collected at a depth of 0.5 m. Water samples were gently (<5 cm Hg) filtered through a GF/F filter and extracted in 90% acetone in the dark at $-20$ °C for 24 h. Chlorophyll-*a* concentrations were then determined fluorometrically using a Tuner designs 10AU fluorometer [27]. Chl-*a* concentrations were expressed in μg Chl-*a* L$^{-1}$. Zooplankton were sampled vertically by towing a modified WP-2 net (diameter 160 mm, mesh size 80 μm) from the bottom of the mesocosm to the surface. Samples were preserved in 70% alcohol. During each sampling event, juvenile fish mortality was assessed. To ensure mixing of the water column, each mesocosm was gently stirred daily for 30 s with a plastic oar. In the laboratory, the total zooplankton abundance at each sampling interval was determined from a $^1/_2$ to $^1/_4$ subsample, obtained by means of a Folsom plankton splitter using a Wild M5A dissecting microscope operating at 100× magnification. Zooplankton abundance data were expressed as Ind L$^{-1}$.

### 2.3. Statistical Analyses

One-way ANOVA with treatment as a fixed factor was employed to determine whether there were any significant differences in the selected physico-chemical and biological variables in the different treatments at the start of the 9-day study. Thereafter, an analysis of covariance (ANCOVA) was run to determine the effect of treatment on the mean total Chl-*a* concentration and the mean total zooplankton abundance. All statistical analyses were performed in R v3.5.1 [28].

## 3. Results

### 3.1. Physico-Chemical Variables

The temperature, salinity and dissolved oxygen concentration were similar across treatments over the duration of the study (Figure 2A–C). The water temperature, salinity and dissolved oxygen concentration ranged from 20.9 to 21.4 °C, from 28.2 to 28.8 and from 7.29 to 7.68 μg L$^{-1}$, respectively. There were no significant differences in the selected physico-chemical variables between treatments at the start of the experiment or between sampling days over the duration of the 9-day experiment ($p > 0.05$ in all cases).

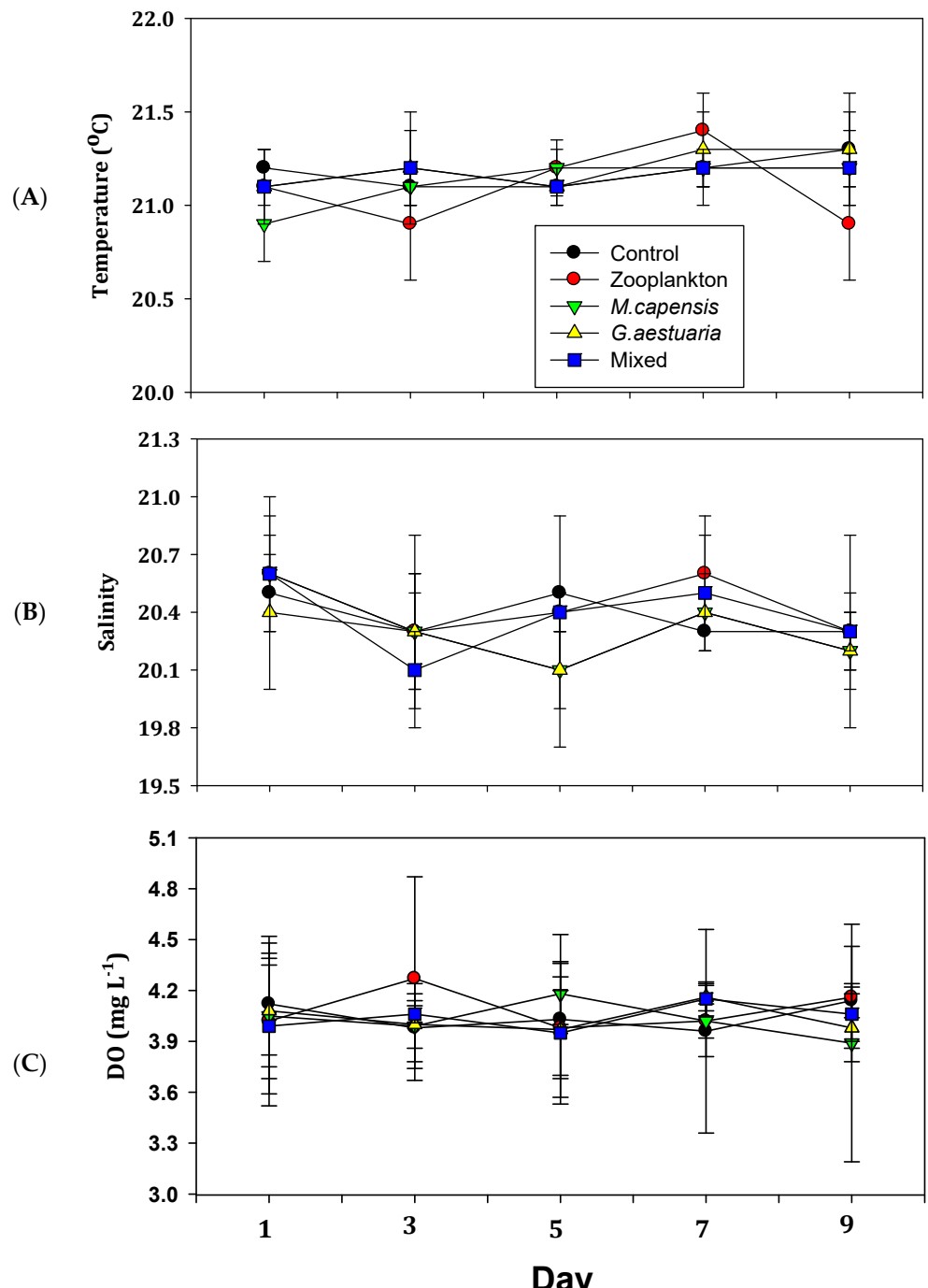

**Figure 2.** Mean (± standard deviation) temperature (**A**), salinity (**B**) and dissolved oxygen concentration (**C**) in different treatments during the 9-day mesocosm study conducted in the middle reach of the temporarily open/closed Kasouga Estuary on the south-east coast of South Africa (n = 3 for each treatment). Note the different scales on the y-axes.

### 3.2. Biological Variables

There were no significant differences in the total chlorophyll-*a* (Chl-*a*) concentrations in the different treatments at the start of the mesocosm experiment (ANOVA; F = 1.35; $p$ = 0.316). By contrast, a post hoc Tukey test conducted after one-way ANOVA indicated that the mean total zooplankton abundance in the control was significantly lower than in the other treatments (F = 42.03; $p < 0.01$). There were no significant differences in the total zooplankton abundances between the zooplankton treatments and the three predator

treatments ($p > 0.05$). Throughout the study, the total zooplankton abundances in the mesocosms were numerically dominated by adult copepods and copepodites of the genera *Pseudodiaptomus*, *Paracartia* and *Oithona*, which accounted for >98% of all zooplankton (data not shown). No juvenile fish mortality was observed in any of the treatments during the 9-day study.

Treatment had a significant effect on the total Chl-*a* concentrations in the different treatments. (F (4, 69) = 35.651 $p < 0.001$). The Tukey post hoc analysis indicated that the mean total Chl-*a* concentration in the control was significantly higher (mean = 0.354) than in the zooplankton (mean = 0.256) and predator treatments. The total Chl-*a* concentration in the zooplankton treatment was significantly lower than in all other treatments (mean = 0.256). There was no significant difference in mean total Chl-*a* concentrations between the *M. capensis* (mean = 0.329), *G. aestuaria* (mean = 0.310) and Mixed (mean = 0.346) treatments.

Treatment, again, had a significant effect on the total zooplankton abundances within the different mesocosms (F (4, 69) = 7.109, $p < 0.05$). The Tukey post hoc analysis indicated that the mean total zooplankton abundance in the control was significantly lower than in all the other treatments (mean = 5.2). Similarly, the mean total zooplankton abundance in the zooplankton treatment was significantly higher (mean = 77.9) than in the control and the apex predator treatments. There was no significant difference in the mean total zooplankton abundances between the *M. capensis* (mean = 38.3), *G. aestuaria* (mean = 43.4) and Mixed treatments (mean = 45.5).

## 4. Discussion

The interactions between predatory species (e.g., cooperation, competition and intra-guild predation), as well as anti-predator responses by prey, can lead to so-called emergent 'multiple predator effects' (MPEs), where prey-consumption rates by multiple predators can have different outcomes which may ultimately affect the strength of trophic cascades [7,11,12]. The current mesocosm study was conducted with the aim of assessing the role of early juvenile diversity in mediating predator–prey outcomes in a shallow-water estuarine ecosystem.

Mesocosms provide a realistic representation of an ecosystem and offer the statistical power of replicated experiments while maintaining many of the key characteristics of the structure and functioning of natural systems [29–31]. In shallow-water ecosystems such as estuaries, benthic–pelagic coupling plays a key role in determining the production, biological structure and food web stability within these systems [32]. The structural complexity conferred by submerged macrophytes has recently been demonstrated to mediate the strength of the interaction between a mysid predator and its prey within a South African estuary [33]. The absence of the benthic community, including the presence of submerged macrophytes, within the mesocosms suggests that the main findings of the study should be viewed with caution, as they do not adequately represent the inherent complexity of the estuarine food web. Nonetheless, the main findings of the study provide insights into the interactions and outcomes of the interactions between juvenile fish and their prey within the water column. The estimates of total Chl-*a* concentration, the total zooplankton abundance and the zooplankton community structure within the mesocosms at the start of the experiment are in the range previously reported in the estuary during the closed phase [2,24,25,34].

Ecological studies have long debated the relative importance of the "bottom up" (resource availability) and the "top down" (biological processes) control of aquatic ecosystems [35,36]. The salinity, temperature and dissolved oxygen concentration were consistent across treatments during the experiment (Figure 2A–C). The significant increase in the total Chl-*a* concentration observed in the absence of grazers (control) suggests that the effect of "bottom up" processes in accounting for the trends in total Chl-*a* concentration in the different treatments can be largely discounted. Studies conducted within the estuary [25,26] and in shallow-water ecosystems worldwide [37,38] have demonstrated the importance of copepods as grazers of phytoplankton production. The significant decline in

total Chl-*a* concentration observed in the zooplankton treatment is, thus, not unexpected and highlights the importance of "top down" processes in structuring the phytoplankton community within the estuary during the study. The role of the copepods in controlling the primary production within the estuary is, however, likely to vary temporally in response to changes in the phytoplankton community structure [34]. During periods of reduced freshwater inflow, the low availability of macronutrients contributes to the phytoplankton community being too small be grazed efficiently by the larger copepods [25,34]. Under these conditions, the microbial loop represents the net sink for phytoplankton production.

The presence of juvenile fish contributed to a significant decline in the total zooplankton abundances, with a concurrent increase in the total Chl-*a* concentrations during the mesocosm experiment ($p < 0.05$; Figure 3A,B). The observed pattern is consistent with the expectations of a trophic cascade [2,4,5] and likely reflects the decreased herbivore activity of the copepods as a result of the predation by the early-life-history fish. Planktivorous fish have been shown to play an important role in linking "bottom up" and "top down" processes through predation and nutrient excretion [36]. It is, therefore, conceivable that the increase in total Chl-*a* concentration observed within the predator treatments could also be attributed, in part, to nutrient enrichment. Indeed, fish excretions have been demonstrated to create hotspots for nutrient cycling in oligotrophic streams [39]. Unfortunately, there were no nutrient data available during the study. It is worth noting that enhanced predator diversity may dampen the strength of a trophic cascade as a result of both direct and indirect interactions between predators [6]. Given the high number of fish species that routinely utilise South African estuaries as nursery areas [23], future investigations should consider the potential impact of predator diversity in mediating the interactions between juvenile fish and zooplankton.

There were no significant differences in the total Chl-*a* concentration and zooplankton abundances in the different predator treatments ($p > 0.05$). This result is in contrast to previous investigations, which have shown that increased predator diversity is generally associated with prey risk reduction [9,11]. Despite the important contribution of *G. aestuaria* and *M. capensis* to the total ichthyofaunal biomass in temperate South African estuaries [15,18,21], virtually nothing is known about the direct and indirect biological interactions between these two species. The juvenile spatial distribution of the two species within South African estuaries broadly overlaps with the maximum densities typically recorded in submerged macrophyte beds in the upper and middle reaches of these systems [15,18]. The juveniles of both species prey mainly on calanoid and cyclopoid copepods, with feeding activity peaking during the daytime [40,41]. The overlap in distribution and diet point to a high degree of intra-guild competition between these two species [42]. Competition for resources between early-life-history fish is typically associated with a decrease in growth rates and increased mortality [43,44]. It is unclear whether competition for resources between the juveniles of the two fish species contributes to their decreased fitness. Alternatively, it is possible that prey availability may be sufficient to sustain the juveniles of both species without any deleterious effects on their fitness, a phenomenon known as superabundance [45].

In contrast to previous studies [6,11], the results of the current investigation indicate that an increase in the diversity of juvenile fish predators did not contribute to a decreased prey risk or the strength of a trophic cascade. The absence of any apparent response suggests that the two juvenile predators act independently of one another [9,13]. The variations in emergent multiple predator effects (i.e., prey risk enhancement or prey risk reduction) are thought to be the result of predators utilising different habitats or employing different foraging strategies [9]. The outcome of the interactions between the two species may, thus, vary according to habitat type or predator life history [46,47]. Further field studies are, therefore, required to better understand the habitat use, trophodynamics and nature of the biological interactions between the juveniles of *G. aesturia* and *M. capensis* within South African estuaries. Predation can promote coexistence among competing prey, thereby enhancing prey diversity [48,49]. Such changes are likely to have far-reaching consequences for the plankton food web structure. Unfortunately, the current investigation

did not consider the impact of the increase in predator diversity on the plankton community composition and diversity within the estuary.

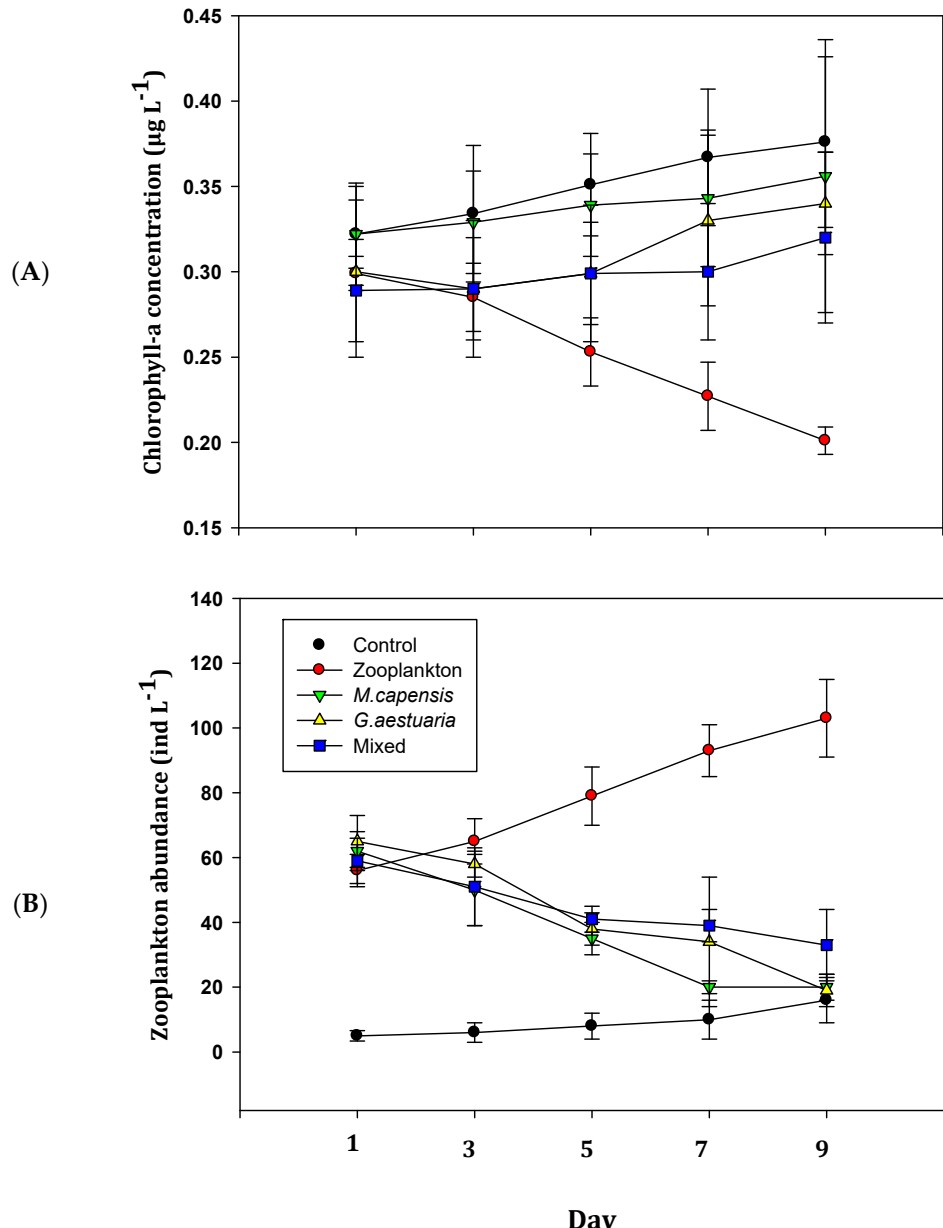

**Figure 3.** Total chlorophyll-*a* concentration (**A**) and total zooplankton abundances (**B**) in the different treatments during the 9-day mesocosm study conducted in the warm temperate, temporarily open/closed Kasouga Estuary, South Africa. Error bars are standard deviation (n = 3 for each treatment). Note the different scales on the y-axes.

**Funding:** This research was funded by Rhodes University, grant number: IFRR10048.

**Institutional Review Board Statement:** All the necessary permits for the collection of the zooplankton and juvenile fish in the experiments were obtained from the Department of Agriculture, Forestry and Fisheries (DAFF), Republic of South Africa (permit reference number: RES2011/46).

**Data Availability Statement:** The data presented in this study are available on request from the corresponding author.

**Acknowledgments:** Funds and facilities for this investigation were obtained from Rhodes University. I would like to thank David Pur-chase for his assistance with the field work.

**Conflicts of Interest:** The author declares no conflict of interest.

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
