# Peer review of "Predator Diversity Does Not Contribute to Increased Prey Risk: Evidence from a Mesocosm Study"

_diversity, doi:10.3390/d14080584_

Round 1

Reviewer 1 Report

Predations plays great roles within aquatic food webs through direct effect on their prey and trophic cascades on lower trophic levels. The manuscript find the combination of two predators does not contribute to increased prey risk based on a mesocosm study. The manuscript is well-written and I would recommend accept after some minor revision.

The main question is the title, as the author compared the predation effect of two single species and the combination on the prey, would “diversity” be a proper title?

Minor points:

L28: comma missed

Line 69: The date when the experiment was conducted should be provided.

Line 78: delete one full stop

Line 79: chlorophyll a, a should be presented in italic.

Line 81: delete one full stop

L102-124: A table is needed here to show the detail in each treatment, e.g. how many fish has been used in each treatment? Did any fish die after the experiment?

L273-274 prey risk enhancement or prey risk enhancement?

Author Response

I would like to thank the reviewer for his/her constructive comments on the manuscript. Where necessary I have revised the manuscript. Below I have listed the responses to the reviewer's comments/suggestions:

  1. L28. As suggested by the reviewer, the comma has been inserted.
  2. L69. As requested by the reviewer, I have now included the date when the experiment was conducted. See page 3, paragraph 3, line 3 of the revised manuscript.
  3. L78. As requested by the reviewer, I have deleted the full stop.
  4. L 79. As requested by the reviewer the "a"_ in Chlorophyll-a has been italicized throughout the revised manuscript. 
  5. L81. The full stop had been deleted from the revised manuscript.
  6. L102-124.  As requested by the reviewer, I have included a sentence in the revised manuscript that now informs the reader of the predator densities in the various treatments. See page 5, paragraph 1, line 8 of the revised manuscript.  Following this, in the original manuscript, I indicated that no predator mortality was observed in the different treatments over the duration of the experiment. See page  7, paragraph 1, line 3 of the revised manuscript.
  7. L273-274. The typographical error has been addressed. "(prey risk enhancement or prey risk enhancement)' now reads, "(prey risk enhancement or prey risk reduction)".  See page   10, paragraph 2, line 5 of the revised manuscript.

Reviewer 2 Report

This paper presents a an elegant experimental study on predator diversity on prey abundance in a river.

A brief discussion would be nice on the relationship between biomass, abndance and species diversity - how are they related in top-down control, for example.

line 9: I do not understand "however", I suggest to delete it

line 12: Southern African? (also in line 44 and later), so you mean your country or the south of Africa?

I understand that comparingsingle-species and two-species cases is like understanding the building blocks of diversity, and from an experimental perspective this is perfectly all right. Yet, from a macroecology perspective one would not call this a "diversity study". I think it would be great to cinvincingly discuss whether adding more and more species would probably follow the same patterns / logic or not.  This needs some guess on the additivity of these effects. This is speculative, I know, but explicit studies (either experimental or theoretical) would be clearly crucial here. I suggest to look at these papers:

https://www.sciencedirect.com/science/article/pii/S0304380020304737

https://esajournals.onlinelibrary.wiley.com/doi/full/10.1890/07-0066.1?casa_token=0y7U5zhjEb8AAAAA%3ASkg675u_gnJ-pKolGQ4juRsYyIj3htJUnc4flcJCcKYLJkrZQNh8c0hPv1Tq-eZBAz0EyFU4BlWYloBEDg

Also, to MPE (line 37), one may think of this classical piece of work:

https://www.science.org/doi/abs/10.1126/science.205.4403.267

line 67: please explain better (and a bit earlier) what is meant by closed and open in the context of a river / mouth. This is not trivial for most readers.

The experimental setup seems to be very well done. Just one question and I know it is not an easy question: what do we know about thesimilarity/variability of the 15 mesocosms at time zero? Are the initial conditions similar? Can we regard them as replicates with good confidence?

Do we know something about the direct interactions (e.g. interference competition but no indirect competition) between these two predatory species?

I am not sure that Table 1 is needed. Maybe you can present these data in a supplement but to ut them in the main text is not justified if the conclucion is just that these are statistically similar.

...and the same for Figure 2, I think.

Clearly not in this paper but it would be superinteresting to repeat this study for 3 species, or for pairs of other species. I would love to grant support for such a research proposal.

Also, to compare different habitats would possibly lead to different results see:

https://link.springer.com/article/10.1007/s42974-021-00066-3

So, you may shortly discuss whether the selected location is representative for the river (or to what extent).

Author Response

I would like to thank the reviewer for his/her constructive comments on the manuscript. Where necessary, I have revised the manuscript to address the concerns/comments of the reviewer. Responses to specific comments/suggestions by the reviewer are listed below.

  1. Line 9. As suggested by the reviewer, I have deleted the word, "However" from the sentence.
  2. Line 12. The term "southern Africa" has been replaced with "South Africa" as this refers to the coast of South Africa only. See page 2, paragraph 2, line 2 of the revised manuscript.
  3. In response to the reviewer's comment, I have included a caveat in the revised manuscript that highlights the need for future studies to consider the potentially important role of the high diversity of predatory juvenile fish in dampening the strength of trophic cascade within the estuary. See page 9, paragraph 2, line 10 of the revised manuscript.
  4. Line 67. In response to the reviewer's comment, I have included a brief description in the revised manuscript that better explains the terms "open vs closed". See page 3,  paragraph 3, line 6 of the revised manuscript.
  5.  There were no significant differences in the selected physico-chemical variables and total chlorophyll-a concentration between the various treatments at the start of the study (See page 6, paragraph 2, line 1 line of the revised manuscript). Given the absence of any significant differences, I am of the opinion that the initial conditions in the different treatments at the start of the experiment were similar.
  6. To the author's knowledge, there is no information available on either the direct or non-direct biological interactions between G.aesturia and M.capensis juveniles in South African estuaries.  
  7. As suggested by the reviewer, Table 1  has been omitted from the revised manuscript. The data are now presented in Figure 2 of the manuscript. 
  8. As suggested by the reviewer, I have now included a caveat in the revised manuscript that informs the reader that the site where the experiment was conducted is likely representative of the estuary during the closed phase of the system. See page 4, paragraph 2, line 3 of the revised manuscript. 
  9. I agree with the comment of the reviewer that the outcome of the interaction between the two juvenile fish species may vary according to habitat type of life history. As a consequence, I have included the following caveat in the revised manuscript'"  The outcome of the interaction between the two juvenile fish species may, therefore, vary according to habitat type or life history [46, 47]."  See page 10, paragraph 2, line 7 of the revised manuscript.
  10. In response to the reviewer's comment, I have included a caveat in the revised manuscript that highlights the need for future studies to consider the impact of increased predator diversity on the plankton community composition and diversity. See page 10, paragraph 2, line 10 of the revised manuscript.